# Global, regional, and national burden of osteoarthritis from 1990 to 2021 and projections to 2035: A cross-sectional study for the Global Burden of Disease Study 2021

**Xiaoming Xie**[1☯], **Kuayue Zhang**[2☯], **Yuan Li**[3☯], **Yulong Li**[4☯], **Xinyi Li**[5], **Yi Lin**[1], **Liangqing Huang**[1], **Guihua Tian**[1]*

**1** Department of Acupuncture, Dongzhimen Hospital Affiliated to Beijing University of Chinese Medicine, Beijing, China, **2** Department of Orthopaedics, Beijing University of Chinese Medicine Third Affiliated Hospital, Beijing, China, **3** Department of Acupuncture, Dongfang Hospital, Beijing University of Chinese Medicine, Beijing, China, **4** Department of Oncology, China-Japan Friendship Hospital, Beijing, China, **5** Department of Electronics, Tsinghua University, Beijing, China.

☯ These authors contributed equally as the first author to this work.
* rosetgh@163.com

## Abstract

### Objective

This study aims to report the trends and cross-national disparities in the burden of osteoarthritis (OA) by region, age, gender, and time from 1990 to 2021, and to further project changes through 2035.

### Methods

In this systematic analysis based on the Global Burden of Disease (GBD) study, population survey data on osteoarthritis from 21 countries/regions and U.S. insurance claims data were used to estimate the prevalence and incidence of OA in 204 countries and regions from 1990 to 2021. The reference case definition for OA was symptomatic and radiographically confirmed osteoarthritis. Studies using definitions other than the reference, such as self-reported OA, were adjusted through a regression model to align with the reference case. The distribution of OA severity was derived from a pooled meta-analysis using the Western Ontario and McMaster Universities Arthritis Index (WOMAC). Final prevalence estimates were multiplied by disability weights to calculate years lived with disability (YLD). An Autoregressive Integrated Moving Average (ARIMA) model was used to forecast the prevalence and incidence of OA through 2035.

### Results

In 2021, approximately 607 million (95%UI 538–671) people worldwide were affected by osteoarthritis, accounting for 7.7% of the global population. Compared to 2020,

**Data availability statement:** In this study, we utilized OA data from the GBD database. All GBD 2021 data are publicly available online (https://vizhub.healthdata.org/gbd-compare/ and https://vizhub.healthdata.org/gbd-results).

**Funding:** This work was supported financially by the National High Level Traditional Chinese Medicine Hospital Clinical Research Funding (DZMG-XZYY-23001).

**Competing interests:** The authors have declared that no competing interests exist.

the age-standardized prevalence of OA among males is projected to increase from 5,763 per 100,000–5,922 per 100,000 by 2036, while the age-standardized prevalence among females is expected to decline slightly from 8,034 per 100,000–7,925 per 100,000. In 2021, the global age-standardized YLD rate for osteoarthritis was 244.5 (95%UI 117.06–493.11), the global age-standardized prevalence rate was 6,967.29 (95%UI 6,180.7–7,686.06), and the global age-standardized incidence rate was 535 (95%UI 472.38–591.97). In 2021, the age-standardized prevalence rate exceeded 5.5% across all regions, ranging from 5,675.8 per 100,000 (95%UI 5,001.76–6,320.8) in Southeast Asia to 8,608.63 per 100,000 (95%UI 7,674.07–9,485.19) in high-income Asia Pacific regions. The knee was the most commonly affected joint. High BMI and metabolic risks are the only two GBD risk factors for osteoarthritis. From 1990 to 2021, the age-standardized prevalence, incidence, and YLD attributable to osteoarthritis have been on the rise, with substantial international variations across indicators. Countries with high socio-demographic index (SDI) bear a disproportionately high burden of OA, and inequalities in the burden of disease due to differences in SDI between countries have been increasing over time.

## Conclusions

As a major public health problem, the overall global burden of OA has shown an upward trend from 1990 to 2019, including an increase in the number of cases and inequalities in distribution across the globe, which has resulted in significant health losses and economic burdens. In addition, SDI-related inequalities between countries are increasing. In this regard, national public health authorities and the World Health Organization (WHO) should work together to improve diagnosis and early treatment rates by strengthening disease awareness and education, as well as strengthening international cooperation, providing necessary medical assistance to less developed regions, and actively exploring new strategies for the prevention and treatment of OA.

## 1. Introduction

Osteoarthritis (OA) is the most common musculoskeletal disorder among middle-aged and elderly populations [1]. It is characterized by pathological changes such as cartilage degeneration, bone remodeling, and osteophyte formation, and its clinical manifestations include joint pain, stiffness, swelling, and functional limitations [2]. As early as 2019, the global number of people with osteoarthritis exceeded 500 million, making OA one of the leading causes of chronic pain and long-term disability in older adults [3]. With the rapid acceleration of global aging and the sharp rise in obesity rates across all age groups, the incidence of OA continues to increase [3], due to the fact that with age, articular cartilage undergoes progressive degeneration due to prolonged mechanical loading and diminished self-repairing ability, which directly increases the risk of OA development; at the same time, obesity exerts excessive mechanical stress on weight-bearing joints (especially the knees), and the adipose

tissue secretes pro-inflammatory cytokines that exacerbate cartilage degeneration and synovial inflammation. By 2020, the global prevalence of OA had risen by 132.2% compared to 1990 [4]. In 2016, the United States spent approximately $80 billion on healthcare for osteoarthritis [4]. As early as 2003, the direct economic burden of OA in Hong Kong ranged from 11,690–40,180 HKD per person annually, with indirect costs ranging from 3,300–6,640 HKD per person annually [5].

However, due to the exclusion of osteoarthritis from global strategies for non-communicable diseases (NCDs), along with the widespread misconception that OA is an inevitable part of aging and lacks effective treatment options, the significant personal, economic, and societal burden of OA has not received sufficient attention [6].

The Global Burden of Disease (GBD), Injuries, and Risk Factor Study aims to provide reliable and up-to-date global, regional, and national estimates on the burden of diseases, injuries, and risk factors by integrating all available data, including published literature, gray literature, survey data, as well as hospital and clinical data [7]. In the area of the global burden of OA, we must acknowledge and appreciate the pioneering work of the GBD 2021 Osteoarthritis Collaborators [3] (published in The Lancet Rheumatology in 2023), which comprehensively analyzes the global burden of osteoarthritis up to the year 2020, and predicts trends up to 2050. Their study provides important insights into the epidemiology of OA, risk factors, and regional variations, and serves as a foundational reference for research on the burden of OA. However, we have to point out that although previous studies have evaluated the burden of OA, they have either focused on specific joint sites [8], single regions [9](such as China), OA related to specific risk factors [10], or provided projections for certain areas [11]. Consequently, findings on OA prevalence vary across studies [12].

Compared to previously published studies on global OA epidemiology, this study employed methods such as trend analysis, decomposition analysis, frontier analysis, and predictive analysis to examine the burden of OA from multiple perspectives, including gender, age, and time. Secondly, Our study explicitly examines SDI-related inequalities in OA burden over time, highlighting how disparities between high- and low-SDI regions have widened since 1990. Thirdly, we identified diverging trends in age standardized prevalence rates between males and females (projected increases for males vs. declines for females by 2035), a finding not explored in the earlier study. Additionally, by incorporating 2021 data, we highlight shifts in OA burden hotspots, such as the rising prominence of high-income Asia Pacific regions (e.g., South Korea) compared to previous emphasis on North America and Western Europe. At last, we conducted a further analysis of OA risk factors. All of these new findings from diverse research approaches will help us to better understand the changing burden of OA and future trends, and will help global public health policy makers to take more proactive measures to combat OA.

## 2. Study design and methods

### 2.1. Data Sources and Methods

The GBD 2021 database provides comprehensive estimates of the prevalence, incidence, years lived with disability (YLD), years of life lost (YLL), and disability-adjusted life years (DALY) at global, regional, and national levels for 204 countries. It covers 369 diseases and injuries, along with 87 risk factors [13]. In this study, we utilized OA data from the GBD database. All GBD 2021 data are publicly available online (https://vizhub.healthdata.org/gbd-compare/ and https://vizhub.healthdata.org/gbd-results). Within the GBD 2021 framework, OA was categorized based on the Western Ontario and McMaster Universities Arthritis Index (WOMAC) [14] into three severity levels: mild, moderate, and severe. DALY is the standardized measure used to quantify the burden of disease, and DALY = YLL + YLD. As the cause-of-death model of GBD assumes no deaths attributable to OA, the DALY for OA is equivalent to YLD. To calculate YLD, the prevalence in each severity group was multiplied by its specific disability weight [15]. We extracted the raw data for OA incidence, prevalence, and YLD by age group and sex at global, regional, and national levels for 204 countries, divided into 21 regions, such as Western Pacific, Central Asia, and Eastern Europe.The OA data we extracted for GBD 2021 included: total OA, knee osteoarthritis, hip osteoarthritis, hand osteoarthritis and other osteoarthritis.

This work has been reported under the STrengthening the Reporting of OBservational studies in Epidemiology (STROBE) standard [16].

## 2.2. Ethical considerations

The University of Washington Institutional Review Board approved a waiver of informed consent for the use of de-identified data in the GBD study.

## 2.3. Case definition

OA was defined as symptomatic OA confirmed by radiographic evaluation, with the Kellgren-Lawrence (KL) grade ranging from II to IV [17–19]. KL grade II is the minimum requirement for diagnosing OA, requiring the presence of pain for at least one month within the past 12 months and the presence of osteophytes on radiographic assessment of the affected joint. Grades III and IV are characterized by osteophytes and joint space narrowing in the affected joint, with grade IV indicating deformity. The primary input data for hip and knee OA models were derived from population-based cross-sectional surveys from around the world and U.S. state-level insurance claims data, captured by International Classification of Diseases (ICD)-9 four- or five-digit codes (beginning with 715) and ICD-10 codes M16 and M17. ICD-10 uses the term "osteoarthritis", replacing "osteoarthrosis" from ICD-9, reflecting the understanding that inflammation is involved in the pathogenesis of OA. GBD 2021 continues to include two new OA categories added in GBD 2019: hand OA and a residual category for OA in other joints (e.g., shoulder and elbow). Consistent with hip and knee osteoarthritis modelling, symptomatic, radiographically confirmed osteoarthritis in any single joint of the hand was used as the reference case definition for hand osteoarthritis, to which alternative case definitions present in the literature were adjusted. Given the paucity of survey data on other osteoarthritis joint sites, US insurance claims data from 2000 to 2016 constituted the sole source of other osteoarthritis data [3].

## 2.4. Descriptive analysis

A descriptive analysis was conducted at global, regional, and national levels to provide a comprehensive overview of the burden of OA. We visually presented the incidence, prevalence, and YLD, including age-standardized rates (ASR), of global OA cases from 1990 to 2021. Furthermore, comparisons of OA cases and ASRs in 1990 and 2021 were made at global, regional (21 GBD regions), and national levels (204 countries/territories).

## 2.5. Trend analysis

Exploring time trends in disease burden is an essential aspect of epidemiology and supports the development of more precise prevention strategies [20]. We used Joinpoint regression analysis to quantify differences in the burden of OA over time and by gender [21]. Joinpoint regression divides overall trends into several segments based on identified breakpoints and calculates the annual percentage change (APC) and 95% confidence intervals (CI) for each segment to evaluate the magnitude of each epidemiological trend. When the average annual percent change (AAPC) and its 95% CI are both above zero, the trend is defined as increasing. Conversely, if the AAPC and its 95% CI are both below zero, the trend is defined as decreasing [22]. Otherwise, the burden is considered relatively stable. The AAPC for the periods 1992–2002, 2002–2012, and 2012–2021 were also estimated by weighting each segment's regression coefficients by the width of each time interval.

## 2.6. Frontier analysis

The Socio-Demographic Index (SDI) was developed by GBD researchers and serves as a composite indicator of country's development level closely related to health outcomes [23]. It measures development level based on lag-distributed

per capita income, fertility rate for the population under 25 years, and average years of education for individuals aged 15 and older [24]. A region with an SDI of 0 represents the lowest theoretical level of development related to health, while an SDI of 1 represents the highest. The 204 countries and regions were divided into five groups based on their SDI: low, low-middle, middle, high-middle, and high [25]. Frontier analysis was conducted to evaluate the ideal YLD levels for osteo-arthritis in each of the 204 countries and regions at their corresponding SDI levels. Through frontier analysis, we identified the countries and regions with the most significant gaps, including: (1) the 15 countries with the largest gap from the fron-tier YLD for OA globally; (2) the five countries with the smallest gap from the frontier value in low SDI regions (SDI < 0.5); and (3) the five countries with the largest gap from the frontier YLD for OA in high SDI regions [26] (SDI > 0.85).

## 2.7. Forecasting OA burden to 2035

The Autoregressive Integrated Moving Average (ARIMA) model is a widely used time series analysis method for fore-casting future values [27]. The ARIMA model captures trends and seasonality in time series data by combining three components: autoregression(AR), integration (I), and moving average (MA). ARIMA parameters are typically expressed as ARIMA(p, d, q), where: p (order of autoregression): indicates how many previous time points are used to predict the current value. The auto regressive component implies that the current value depends on a linear combination of its previ-ous p values; d (degree of difference): represents how many times the data needs to be differenced to become stationary. Differencing refers to calculating the difference between adjacent observations to remove trends or seasonality in the time series; q (order of moving average): indicates how many previous error terms are used to predict the current value. The moving average component implies that the current value depends on a linear combination of the previous q error terms [28]. Mathematically, an AR(p) model can be expressed as: $y_t = \alpha_0 + \alpha_1 y_{t-1} + \alpha_2 y_{t-2} + \cdots + \alpha_p y_{t-p} + \epsilon_t$, where $y_t$ represents the observed value at time t, $\alpha_i$ are the regression coefficients, and $\epsilon_t$ is the random error term; If the data is non-stationary, differencing can be applied to remove the trend. For example, first-order differencing (d = 1) calculates the difference between the current value and the previous value: $\Delta y_t = y_t - y_{t-1}$. Repeated differencing operations can eliminate more complex trends; The MA(q) model can be represented as: $y_t = \mu + \epsilon_t + \theta_1 \epsilon_{t-1} + \theta_2 \epsilon_{t-2} + \cdots + \theta_q \epsilon_{t-q}$, where $\epsilon_t$ is the error term and $\theta_i$ are the coefficients of the error terms [29].

## 2.8. Risk factor analysis

Bayesian meta-regressions of the adjusted data were run using DisMod-MR 2.1, 17 an age-integrating Bayesian meta-regression log-normal disease model with a mixed-effects geographical cascade. The meta-regression was a combination of a meta-analysis to pool data points with weighted averages to include and reconcile heterogeneous data, and a regres-sion to include known associations between several variables (e.g., osteoarthritis prevalence and BMI or age). Fixed effects included sex and country-level covariates (e.g., BMI). Nested random effects were calculated for each super-region, region, and country [30].

Within GBD, DisMod-MR 2.1 uses BMI as a fixed covariate solely to improve OA prevalence estimation by leveraging known BMI-OA associations when primary data are sparse (Flaxman [31] et al., 2015). In a separate Comparative Risk Assessment, each metabolic component-high fasting plasma glucose, high systolic blood pressure, high BMI, low bone mineral density, high LDL cholesterol, impaired kidney function-is assigned its own relative-risk curve and theoretical min-imum risk exposure level (TMREL). The six resulting PAFs are then combined multiplicatively to produce total metabolic risk, thereby avoiding double-counting of BMI effects (GBD 2019 Risk Factors Collaborators [32], 2020).

Our analysis, aligned with the Global Burden of Disease (GBD) 2021 framework, focused on two risk factors: high BMI and metabolic risks. The latter encompasses high fasting plasma glucose, high systolic blood pressure, low bone mineral density, high LDL cholesterol, and impaired kidney function. These factors were selected based on their mechanistic and epidemiological links to OA. For instance, high BMI is a well-established causal driver of OA, supported by meta-analyses (Blagojevic M [33], et al, 2010), longitudinal cohort studies (Felson D.T [34], et al, 1988)and Mendelian randomization

studies (Zengini E [35], et al, 2018). Similarly, metabolic risks were included due to their roles in systemic inflammation, cartilage degradation, and subchondral bone remodeling, with evidence from longitudinal cohorts (Hoeven TA [36], et al, 2013; Clockaerts S [37], et al, 2010; Kim CS [38], et al, 2019).

All in all, The inclusion of high BMI and metabolic risks reflects their robust causal associations with OA. For example, a meta-analysis of 63 studies confirmed that obesity (BMI ≥ 30 kg/m²) increases knee OA risk by 2.63-fold (Blagojevic M [33], et al, 2010), while hyperglycemia (Louati K [39], et al, 2015) and hypertension (Wang T [40], et al, 2017) are independently linked to OA progression.

### 2.9. Statistical analysis

Joinpoint regression analysis was performed using the Joinpoint Regression Program V5.0.2 provided by the U.S. National Cancer Institute Surveillance Research Program. All analyses and visualizations in this study were conducted using the World Health Organization (WHO) Health Equity Assessment Toolkit and R software (V.4.3.2).

## 3. Results

### 3.1. Descriptive analysis of the OA burden at global, regional, and national levels

In 1990, 4.8% of the global population had osteoarthritis, amounting to approximately 256 million (95%UI 232–282) people. By 2021, the estimated global prevalence had risen to 7.7%, with around 607 million (95%UI 538–671) people affected. The number of cases has steadily increased across all age groups over the following decades. Table 1 and S1 Table presents the global and regional incidence, prevalence, YLD data, and corresponding ASR for OA in 2021. Among the GBD regions, the highest age-standardized prevalence rates were observed in high-income Asia Pacific (8,608.63 per 100,000, 95%UI 7,674.07–9,485.19), high-income North America (8,421.62 per 100,000, 95%UI 7,534.98–9,282.03), and Eastern Europe (7,906.11 per 100,000, 95%UI 6,954.04–8,880.09). In contrast, the lowest age-standardized prevalence rates were recorded in Southeast Asia (5,675.8 per 100,000, 95%UI 5,001.76–6,320.89), Eastern Sub-Saharan Africa (5,829.96 per 100,000, 95%UI 5,160.63–6,476.62), and Central Sub-Saharan Africa (5,940.49 per 100,000, 95%UI 5,268.27–6,589.54)

The number of OA patients of different ages and genders worldwide in 2021, as well as the standardized prevalence and incidence rates, are shown in (Fig 1). OA is more prevalent among individuals aged 35 and older, with higher and rapidly increasing prevalence observed in the 35–90 age group. The peak prevalence for females occurs between the ages of 65 and 69, while for males, it peaks between the ages of 55 and 59. A similar trend is observed in incidence rates, with a sharp increase after the age of 30. The peak incidence for both men and women is between the ages of 50 and 54. Across all estimated years, the incidence and prevalence rate of osteoarthritis has been consistently higher in females than in males.

The sex-specific, age-standardized incidence and prevalence rates of OA have fluctuated over the calendar years. While the number of OA cases and new incidences showed a significant upward trend, the ASR for prevalence and incidence displayed only a slight increase from 1990 to 2021. However, the values for females were consistently much higher than those for males, as illustrated in S1 Fig.

In all GBD regions, knee osteoarthritis is the largest contributor to the age-standardized prevalence rate of mixed osteoarthritis, with the exception of Central Asia and Eastern Europe, where hand osteoarthritis is the predominant contributor (as shown in S2 Fig). The proportion of knee osteoarthritis ranges from 34.9% in Central Asia to 66.4% in East Asia. Hip osteoarthritis contributes the least to OA prevalence across all regions, except for high-income North America and Western Europe, where its contribution is similar to or slightly lower than that of other types of OA. The proportion of hip osteoarthritis ranges from 3.3% in East Asia to 9.5% in high-income North America. The contribution of hand osteoarthritis varies by region, with the lowest contribution in East Asia (21%) and the highest in Central Asia (50.3%).

**Table 1. The ASR of prevalence, incidence and YLDs of OA in 1990 and 2021 for both sexes by GBD regions.**

| | 1990 | | | 2021 | | |
|---|---|---|---|---|---|---|
| | ASR of YLDs per 100 000 | ASR of Prevalence per 100 000 | ASR of Incidence per 100 000 | ASR of YLDs per 100 000 | ASR of Prevalence per 100 000 | ASR of Incidence per 100 000 |
| Global | 222.8 (106.65,450.29) | 6393.12 (5683.2,7059.53) | 489.78 (433.1,541.51) | 244.5 (117.06,493.11) | 6967.29 (6180.7,7686.06) | 535 (472.38,591.97) |
| East Asia | 211 (102.08,424.58) | 6157.54 (5425.37,6866.85) | 487.3 (428.32,544.04) | 245.04 (117.45,492.41) | 7036.1 (6216.29,7835.76) | 554.47 (486.91,619.37) |
| Central Asia | 215.8 (104.18,434.3) | 6143.82 (5368.48,6965.53) | 444.99 (392.55,498.51) | 249.29 (119.63,500.56) | 7034.89 (6120.08,8010.15) | 504.46 (442.21,565.75) |
| Southeast Asia | 163.3 (78.14,329.58) | 4796.58 (4256.53,5377.23) | 376.04 (332.26,418.37) | 196.18 (93.56,393.42) | 5675.8 (5001.76,6320.89) | 437.13 (386.13,485.01) |
| Oceania | 192.33 (92.49,385.92) | 5637.24 (5022.89,6261.36) | 441.66 (389.8,491.69) | 212.87 (102.99,428.52) | 6196.48 (5474.55,6895.02) | 480.95 (423.12,536.36) |
| Eastern Europe | 265.83 (126.81,540.87) | 7541.08 (6611.08,8496.07) | 550.43 (484.03,614.18) | 280.78 (134.37,567.03) | 7906.11 (6954.04,8880.09) | 584.97 (515.25,651.42) |
| Western Europe | 238.56 (115.19,479.88) | 6736.67 (6071.81,7425.01) | 521.51 (465.66,578.73) | 253.58 (123.06,510.55) | 7113.44 (6407.11,7867.1) | 557.66 (497.27,618.53) |
| Southern Latin America | 247.99 (118.32,499.99) | 7001.84 (6250.94,7759.15) | 540.18 (478.83,600.89) | 273.53 (131.22,548.73) | 7669.24 (6896.46,8466.3) | 596.27 (530.39,660.42) |
| High-income North America | 286.25 (137.18,576.08) | 7987.16 (7188.9,8824.97) | 605.72 (535.69,670.62) | 300.89 (144.87,606.97) | 8421.62 (7534.98,9282.03) | 646.38 (572.29,715.37) |
| Caribbean | 228.48 (109.52,461.2) | 6514.84 (5762.16,7198) | 513.03 (455.98,570.5) | 251.06 (120.09,508.25) | 7134.56 (6327.26,7876.59) | 555.77 (493.24,617.51) |
| Andean Latin America | 231.91 (111.18,466.16) | 6602.95 (5861.3,7291.11) | 519.51 (461.53,577.69) | 260.94 (125.19,526.82) | 7370.44 (6552.1,8123.13) | 578.18 (511.33,641.1) |
| Central Latin America | 231.66 (110.81,468.03) | 6661.13 (5900.48,7353.18) | 527.96 (468.13,587.05) | 264.58 (126.15,535.65) | 7499.49 (6635.38,8259.93) | 589.49 (521.45,652.52) |
| Tropical Latin America | 228.15 (109.07,460.14) | 6604.05 (5863.57,7307.66) | 527.59 (467.44,585.25) | 259.93 (124.74,524.63) | 7424.65 (6582.79,8241) | 589.12 (521.39,650.6) |
| North Africa and Middle East | 183.41 (87.76,371.84) | 5362.22 (4751.55,5979.22) | 424.31 (375.36,470.67) | 215.92 (103.37,437.62) | 6265.22 (5572.94,6946.23) | 488.31 (433.7,542.33) |
| Central Europe | 218.64 (104.72,441.82) | 6276.97 (5554.86,7001.94) | 474.27 (419.37,525.88) | 245.41 (117.63,496.21) | 6948.51 (6129.15,7752.72) | 522.05 (460.81,580.35) |
| Australasia | 254.48 (123.19,513.81) | 7195.28 (6466.74,7950.86) | 555.78 (493.55,617.22) | 283.38 (139.24,577.97) | 7917.6 (7098.38,8735.71) | 620.09 (550.76,686.53) |
| South Asia | 181.92 (87.8,368.21) | 5407.04 (4798.72,5985.59) | 430.77 (382.03,477.59) | 216.9 (104,438.04) | 6326.13 (5612.39,7009.64) | 495.01 (436.64,548.03) |
| High-income Asia Pacific | 291.1 (139.9,587.96) | 8071.98 (7169.35,8905.87) | 641.16 (568.18,707.78) | 314.98 (150.55,636.77) | 8608.63 (7674.07,9485.19) | 682.07 (606.06,752.84) |
| Central Sub-Saharan Africa | 191.36 (91.09,387.34) | 5622.81 (4965.8,6260.93) | 441.86 (390.35,491.24) | 204.3 (97.78,412.23) | 5940.49 (5268.27,6589.54) | 463.08 (409.68,515.24) |
| Eastern Sub-Saharan Africa | 173.52 (83.3,351.43) | 5113.71 (4544.28,5704.44) | 411.5 (363.2,457.47) | 200.97 (96.11,405.27) | 5829.96 (5160.63,6476.62) | 461.02 (407.42,509.93) |
| Southern Sub-Saharan Africa | 229.15 (110.1,460.99) | 6559.8 (5794.82,7289.01) | 513.31 (454.83,568.81) | 249.45 (120.36,500.63) | 7161.23 (6333.34,7951.33) | 557.24 (493.49,618.18) |
| Western Sub-Saharan Africa | 187.48 (90.32,378.77) | 5494.22 (4872.02,6124.65) | 439.61 (387.27,489.07) | 210.09 (101.11,424.67) | 6075.81 (5385.72,6757.27) | 483.84 (427.08,536.68) |

Abbreviations: YLDs = years lived with disability, ASR = Age-standardised rate, OA = osteoarthritis, GBD = Global Burden of Disease.

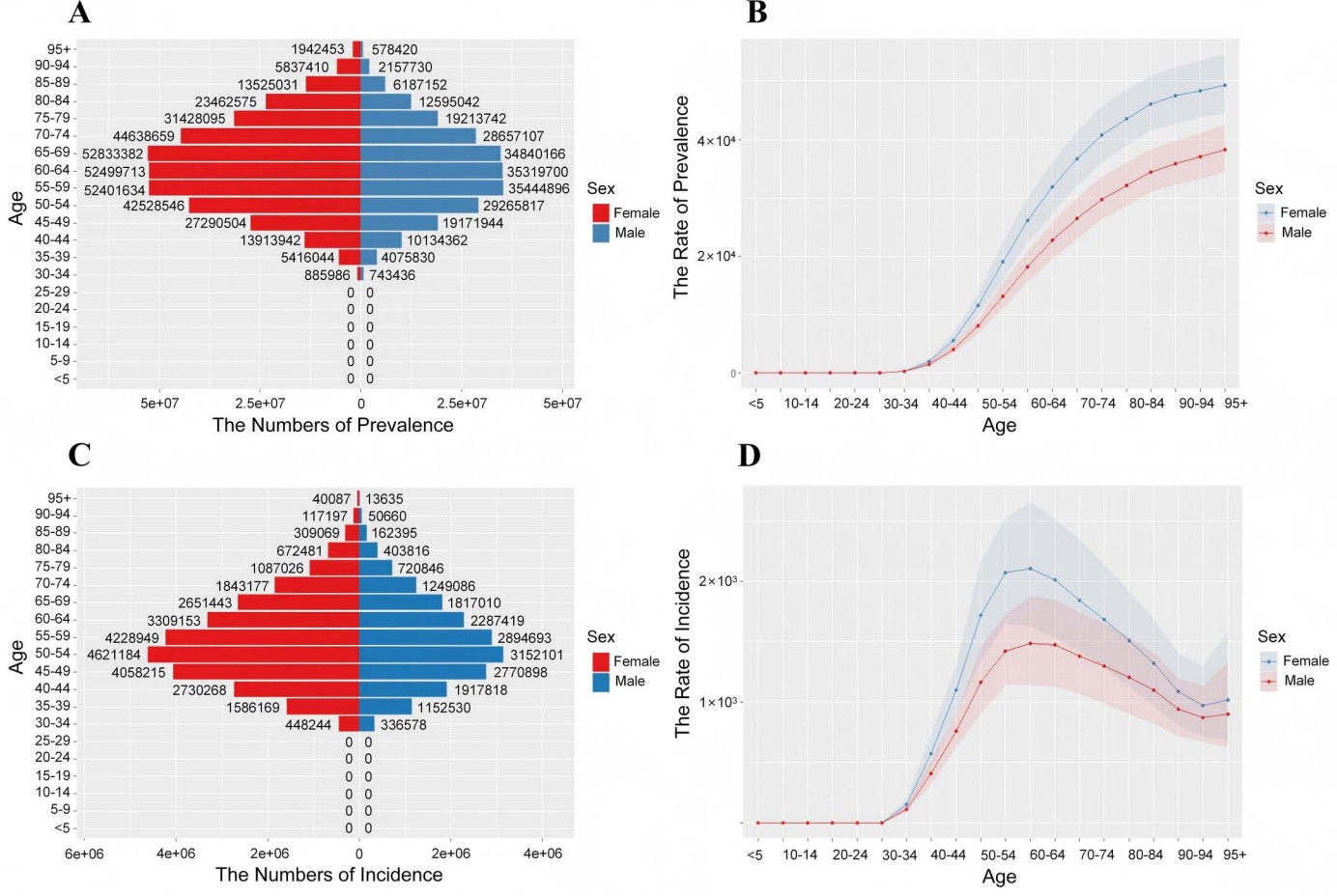

**Fig 1. Age-specific numbers and age-standardized prevalence and incidence rates of OA globally, 2021.** (A) Age-specific prevalence number. (B) Age-standardized prevalence rate. (C) Age-specific incidence number. (D) Age-standardized incidence rate.

Globally, South Korea had the highest age-standardized prevalence rate of OA (8,997.39 per 100,000, 95%UI 8,082.99–9,897.80), followed by Brunei (8,815.59 per 100,000, 95%UI 7,892.91–9,708.06), Singapore (8,795.59 per 100,000, 95%UI 7,834.77–9,662.12), the United States (8,686.57 per 100,000, 95%UI 7,789.66–9,568.34), and Japan (8,442.68 per 100,000, 95%UI 7,514.97–9,306.51). In terms of incidence, South Korea also had the highest age-standardized incidence rate (701.23 per 100,000, 95%UI 625.36–776.78), followed by Brunei (686.67 per 100,000, 95%UI 609.40–759.01), Singapore (685.67 per 100,000, 95%UI 606.53–760.52), Japan (671.40 per 100,000, 95%UI 594.22–740.18), and the United States (668.49 per 100,000, 95%UI 591.69–739.57). As for age-standardized YLD, South Korea ranked highest (327.14 per 100,000, 95%UI 157.44–662.81), followed by Singapore (323.81 per 100,000, 95%UI 156.07–648.84), Brunei (319.00 per 100,000, 95%UI 152.75–642.40), the United States (310.78 per 100,000, 95%UI 149.66–627.25), and Japan (309.47 per 100,000, 95%UI 147.61–625.16), as shown in Fig 2.

### 3.2. Temporal trends in the OA burden from joinpoint regression analysis

The results of the Joinpoint regression analysis on the trends in OA burden are shown in (Fig 3) and S2 Table. We found that the incidence of OA in males showed a significant upward trend during the periods of 2000–2005 (APC = +0.56, 95%

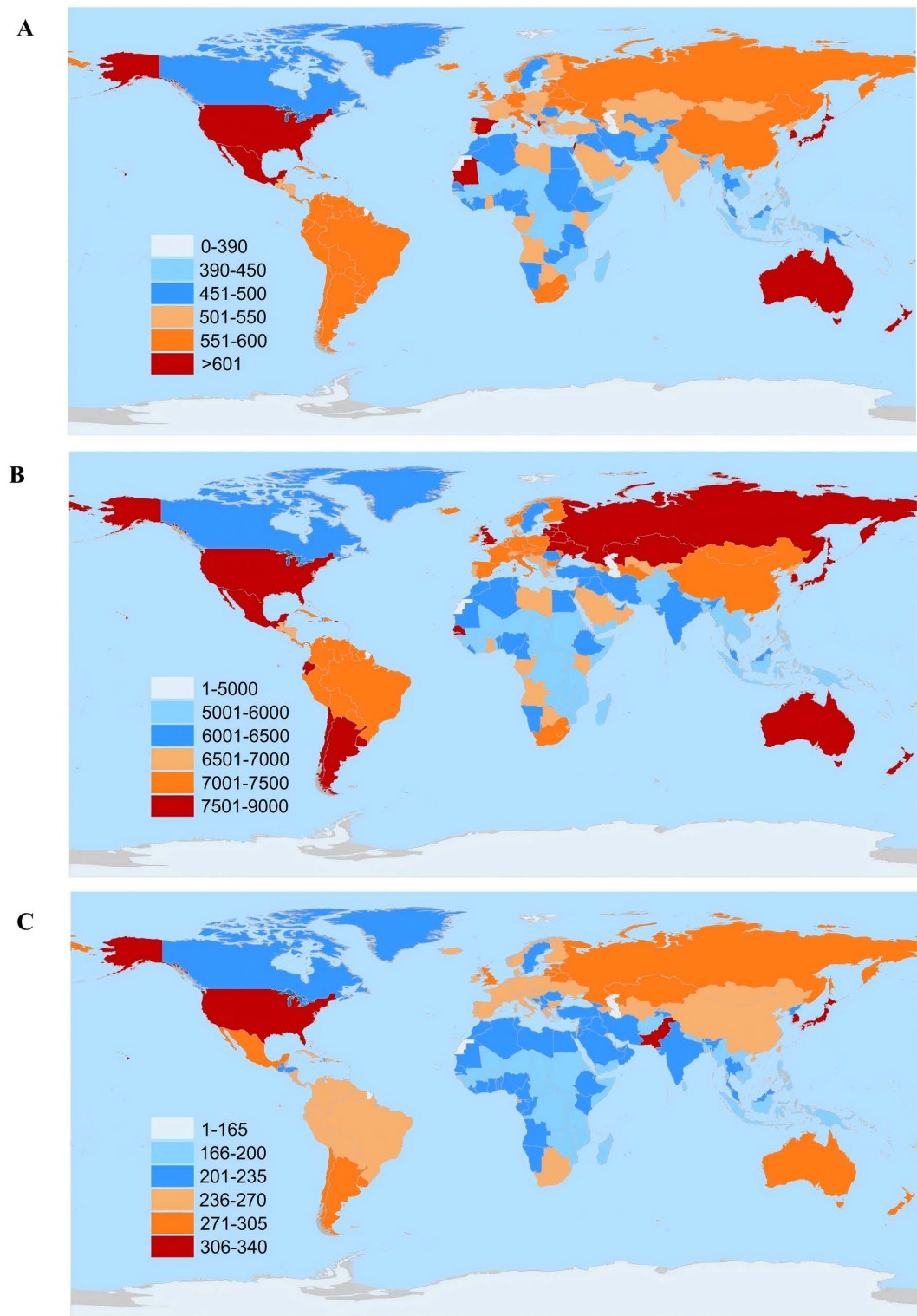

**Fig 2. Age-standardised incidence (A), prevalence (B) and YLDs (C) per 100 000 of total osteoarthritis by country for male and female sexes combined in 2021.** Note: This diagram was drawn using R programming software and is the original work of the authors of this article, which permits anyone to distribute, reproduce, adapt or use it for commercial purposes.

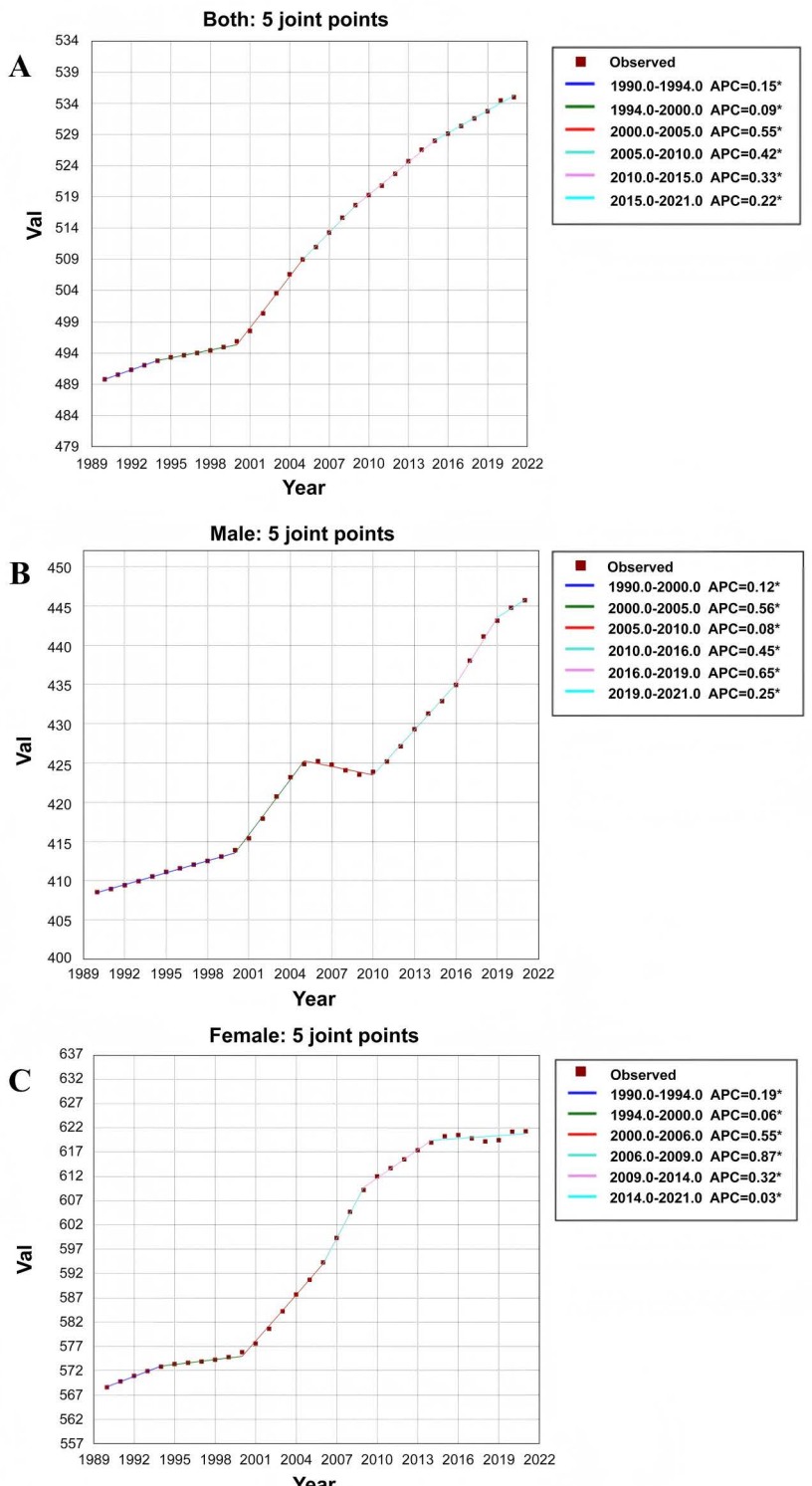

**Fig 3. Joinpoint regression analysis of the sex-specific age-standardized incidence rate for OA globally from 1990 to 2021.** (A) Age-standardized incidence rate for both sexes.(B) Age-standardized incidence rate for male. (C) Age-standardized incidence rate for female.

CI = 0.54–0.59), 2010–2016 (APC = +0.45, 95% CI = 0.43–0.47), and 2016–2019 (APC = +0.65, 95% CI = 0.57–0.72). For females, the incidence of OA experienced a significant increase between 2000 and 2006 (APC = +0.55, 95% CI = 0.49–0.60) and from 2006 to 2009 (APC = +0.87, 95% CI = 0.64–1.11).

From 1990 to 2021, the overall incidence of OA in both males and females showed a continuous increasing trend, although there was a slight decline in male OA incidence between 2005 and 2010 (APC = -0.08, 95% CI = -0.11 to -0.06). Additionally, the average annual percent change (AAPC) in male OA incidence from 1990 to 2021 was +0.28 (95% CI = 0.27–0.29), while the AAPC for female OA incidence was +0.28 (95% CI = 0.25–0.31).

### 3.3. Frontier analysis of the association between ideal OA YLD and SDI

To explore the ideal scenario for controlling disease burden under the corresponding SDI conditions for each country/region, a frontier analysis was conducted (S3 Fig). In the results of the frontier analysis, the five countries/regions closest to the frontier fit line in lower SDI countries/regions are marked in blue, while the five countries/regions farthest from the frontier fit line in higher SDI countries/regions are marked in red. Among all countries, the 15 countries/regions farthest from the frontier fit line are marked in black.

For YLD attributable to OA, the countries/regions farthest from the frontier line in higher SDI areas include South Korea, the high-income Asia Pacific region, the United States, Japan, and the high-income North American region. In contrast, the countries/regions closest to the frontier line in lower SDI areas include the Lao People's Democratic Republic, the Central African Republic, Burkina Faso, the Democratic Republic of the Congo, and Tanzania.

### 3.4. Risk factor analysis for OA

Within the GBD framework, metabolic risk includes: high blood glucose, high blood pressure, low bone density, high LDL cholesterol, high BMI, and impaired kidney function. After we included all the above factors in our analysis, according to the output of GBD (https://vizhub.healthdata.org/gbd-results/): high BMI and total metabolic risk were the only two risk factors.

In 1990, high BMI was responsible for 35.97 (95%UI -3.14 to 102.31) global age-standardized YLDs attributable to OA. By 2021, high BMI was responsible for 50.59 (95%UI -4.81 to 141.34) global age-standardized YLDs attributable to OA. In 2021, the highest burden of age-standardised YLDs for osteoarthritis due to high BMI was in high-income North America and the lowest in South Asia. The burden of age-standardised YLDs for osteoarthritis attributable to high BMI in 1990 vs 2021 can be found in S4 Fig.

### 3.5. Forecast analysis of the OA burden to 2035

The predicted ASR of prevalence and predicted ASR of YLDs for OA through 2035 are shown in (Fig 4). Globally, with the exception of an increase in the age-standardized prevalence rate for males, the ASRs for both OA prevalence and YLDs are expected to decline annually until 2036. Specifically, the age-standardized prevalence rate for males is projected to increase from 5,763 per 100,000 in 2020–5,922 per 100,000 in 2036, while the age-standardized YLDs for males are expected to decrease from 200.23 per 100,000 in 2020 to 195.34 per 100,000 in 2036. For females, the age-standardized prevalence rate is predicted to decrease from 8,034 per 100,000 in 2020–7,925 per 100,000 in 2036, and the age-standardized YLDs for females are expected to decline from 283.79 per 100,000 in 2020 to 281.76 per 100,000 in 2036.

## 4. Discussion

This study provides data on the incidence, prevalence, and YLD of OA at global, regional, and national levels from 1990 to 2021, presenting a comprehensive assessment based on trend analysis, frontier analysis, decomposition analysis, and predictive analysis. It reveals a continuous rise in the global OA burden, with the fastest growth observed between 2000 and 2010. Decomposition analysis by affected joint sites indicates that knee osteoarthritis (KOA) is the largest contributor

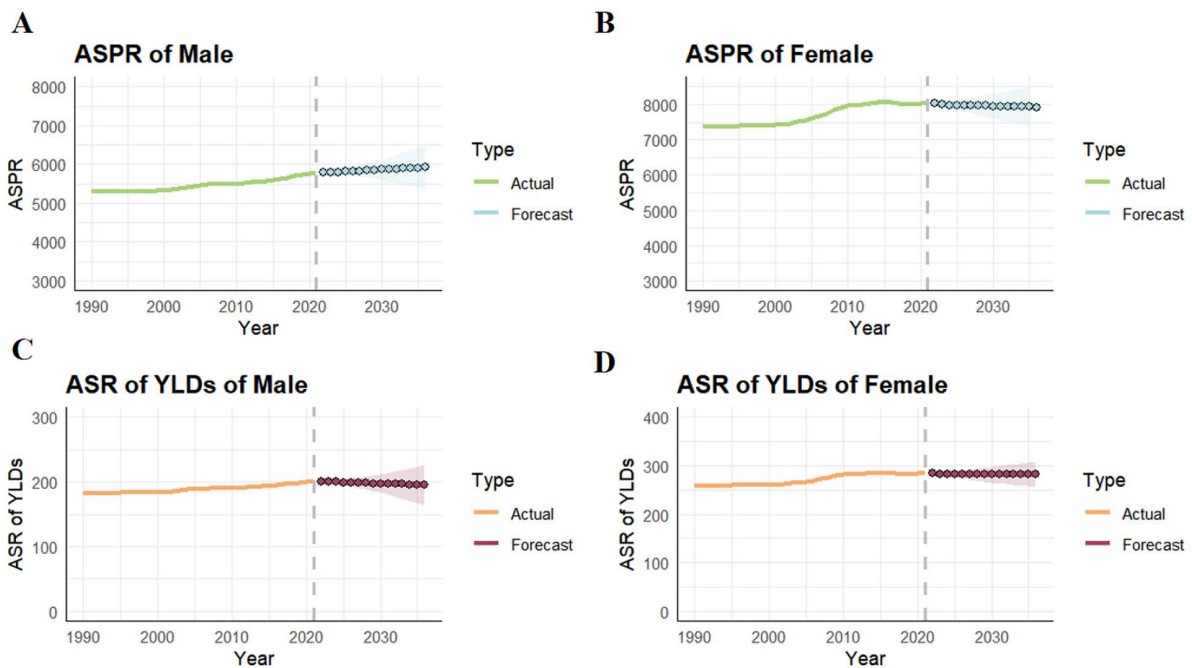

**Fig 4. (A) The predicted ASPR of male to 2035; (B) the predicted ASPR of female to 2035; (C) The predicted ASR of YLDs of male to 2035; (D) the predicted ASR of YLDs of female to 2035.** Abbreviations: ASR, age-standardized rate; OA, osteoarthritis; ASPR, age-standardized rate of prevalence.

to the age-standardized prevalence rate of OA. Although future projections suggest that the ASR for incidence, prevalence, and YLD may slightly decline by 2035, the total number of OA cases will continue to rise due to population aging and growth, leading to a significant public health burden. Managing and controlling OA in the future will pose major challenges, requiring the allocation of healthcare resources and the development of intervention strategies in a scientifically equitable manner.

In 2021, the global OA burden was substantial, with 607 million prevalent cases, 46.63 million new cases, and 21.3 million YLDs. East Asia had the highest number of OA cases (158,285,424, 95%UI 139,469,183–176,824,968), reflecting the region's large population base and accelerated aging. The high-income Asia Pacific region had the highest global age-standardized prevalence rate at 8,608.63 (95%UI 7,674.07–9,485.19). South Korea, facing severe population aging, ranked first globally with an ASR of YLDs at 244.5 (95%UI 117.06–493.11), an ASR of prevalence at 6,967.29 (95%UI 6,180.7–7,686.06), and an ASR of incidence at 535 (95%UI 472.38–591.97). China had the highest number of OA cases in 2021, which is related to its large population. We found that the region with the highest age-standardised prevalence of OA was the high-income Asia-Pacific region, rather than the previously high-income North America; at the national level, the country with the highest age-standardised prevalence of OA changed from the United States to the Republic of Korea, and the Asia-Pacific region of Singapore, Brunei, and Japan had a higher burden of OA, suggesting a significant geographic shift of the burden of disease for OA. Japan is known to have one of the world's highest levels of population aging, having entered a super-ageing society by 2007. As of 2024, the number of elderly people aged 65 years and older in Japan has reached 36.25 million, accounting for 29.3% of the total population [41]; South Korea has the lowest fertility rate in the world, and the trend of population aging is increasing, and it is expected that it will overtake Japan to become the most seriously aging country in 2044 [42]; in addition, China has already reached a demographic inflection point in 2022, and the number of deaths in China was more than the number of births in 2022, with negative population growth

occurring for the first time [43]. Together, these changes illustrate the wide variation in OA burdens and trends across countries and regions, which underscores the importance of flexible, region-specific public health policies, and reminds health policymakers around the globe of the urgency of proactively addressing population ageing [44].

Globally, the incidence, prevalence, and YLD of OA continuously increased from 1990 to 2021, intensifying the global burden. With rising aging and obesity rates, the high incidence of OA places tremendous pressure on healthcare systems and economies worldwide. OA not only incurs direct economic costs (e.g., for medication or surgical interventions) and indirect costs (e.g., lost workdays, premature death) but also results in personal costs (e.g., chronic pain, fatigue, limited mobility). The medical expenditure associated with OA accounts for 1%~2.5% of GDP [45], underscoring the significant economic burden posed by the disease. Therefore, managing and preventing OA effectively will become a crucial global public health issue in the coming decades. Countries must invest in early intervention, disease management, and technological innovation while optimizing resource allocation to ensure timely and effective treatment and care for patients [46].

When analyzing overall trends in multiple segments, we found that from 2000 to 2005, the ASRs for incidence, prevalence, and YLD of OA increased most significantly, marking this period as the fastest-growing phase of the disease burden. Similarly, a study based on GBD 2015 data showed that the rate of increase in YLD due to OA was notably higher before 2005 than after. This might be attributed to the global population aging and the obesity epidemic [47]. Changes in major risk factors for disease onset and progression, along with how governments and individuals respond to these factors, may together influence the time trends of disease burden. Globally, aging and obesity are the primary risk factors for OA, particularly in the early 21st century, when the global burden of OA worsened due to accelerated aging and rising obesity rates. This trend reflects the impact of demographic and lifestyle changes on global public health. To address this trend, the United Nations and the World Health Organization launched a series of global policies in the early 2000s, focusing on addressing population aging and obesity. Aging policies aimed to delay the onset of age-related diseases and promote healthy aging, while anti-obesity policies sought to curb rising global obesity rates through improved diets and increased physical activity. These policies may have contributed to slowing the growth of the OA burden after 2005 and increased global attention to OA management.

When classifying OA by affected joints, KOA emerged as the largest contributor to the total age-standardized prevalence rate of OA, with its proportion ranging from 34.9% to 66.4% across different GBD regions. KOA, therefore, warrants more medical attention and research. The burden of OA is significantly higher in women than in men [48], calling for more effective prevention and management measures tailored to women. Women's unique physiological characteristics, such as hormonal and reproductive factors, significantly affect their OA risk, especially the decline in estrogen levels, which exacerbates cartilage damage and bone loss, increasing the likelihood of OA [49]. Additionally, during physiological stages such as menarche, pregnancy, and menopause, women often experience weight gain accompanied by systemic inflammation, further elevating their OA risk. Taking these factors into account, there is an urgent need to strengthen our focus on women's joint health, to pay more attention to protecting women's health rights and interests, and to promote preventive strategies and interventions to address this growing public health problem.

Musculoskeletal pain is the primary patient-reported outcome in patients with OA and a major cause of disability and functional limitations [50]. Studies have shown that weight loss can reduce many of the symptoms of osteoarthritis, including pain [51]. In the analysis of risk factors, high BMI and metabolic risks emerged as the only two GBD risk factors for OA. Compared to NSAIDs, weight control is more convenient, accessible, and safer, so it is significant that we incorporate more effective weight control into prevention policies at global, regional, and national levels [52]. It is worth noting that the GBD database includes only 88 common disease risk factors, which inevitably limits the comprehensive exploration of OA risk factors to some extent. Furthermore, evidence indicates that patterns of disease and exposure or relative risk functions for exposure and outcome sometimes differ from expert consensus. This discrepancy arises because the GBD study is committed to a rules-based approach to evidence synthesis, which may lead to findings that diverge from other assessments that rely more heavily on expert opinions. Additionally, data collection for the GBD database is closely

tied to the level of economic development in different regions. For many countries in Asia, Africa, and Latin America, the data included in the analysis often lack key risk factors such as physical activity levels, vitamin intake, and analgesic use. Therefore, the conclusion that "high BMI and metabolic risk are the only two significant risk factors for OA" should be approached with caution. However, with the continued expansion of the GBD database and updates to statistical models, there is hope for exploring a broader range of OA risk factors in the future. This progress will facilitate a more comprehensive and systematic understanding of the burden of OA.

Although high-SDI countries are generally expected to have better healthcare systems and lower disease burdens, the burden of OA is disproportionately concentrated in these countries [53] (e.g., South Korea, the United States, Japan). Previous research [54] has shown a positive correlation between OA burden and SDI level, suggesting that OA may become one of the most prevalent diseases in high-income countries. This phenomenon is partly due to population aging and obesity, both of which are key risk factors for OA. Furthermore, the lack of an effective cure for OA continues to drive its rising prevalence. In addition, diagnostic bias stemming from unequal access to healthcare services exacerbates this trend. Therefore, to effectively reduce the OA burden, countries must implement targeted policies and allocate resources based on SDI levels. High-SDI countries should focus on early diagnosis and management strategies, while low-SDI countries need more medical assistance [55]. This call for action requires global cooperation to address the growing inequalities in OA burden.

Notably, although the ASRs for OA incidence, prevalence, and YLDs are expected to decline annually by 2035 (with male age-standardized YLDs projected to decrease from 200.23 per 100,000 in 2020 to 195.34 per 100,000 in 2036, and female age-standardized YLDs projected to decrease from 283.79 per 100,000 in 2020 to 281.76 per 100,000 in 2036), the number of cases for all three indicators is expected to increase. This indicates a heavy disease burden and new challenges in controlling and managing OA.

It is well established that aging and obesity are closely associated with osteoarthritis (OA). With advancing age, the human skeletal and joint systems undergo degenerative changes, including cartilage wear, ligament laxity, and reduced bone density, making older adults more susceptible to OA. Aging exacerbates the burden on weight-bearing joints, particularly the knees and hips. Therefore, older adults should engage in moderate exercise to maintain good joint mobility and muscle strength, thereby preventing joint stiffness. Low-impact aerobic exercises, such as swimming and cycling, as well as strength-training exercises, are particularly recommended [56].Obesity is an independent and significant risk factor for OA. Excessive body weight increases the mechanical load on joints, especially the knees and hips. Additionally, adipose tissue secretes inflammatory factors that can exacerbate cartilage damage and accelerate the onset and progression of OA. Thus, weight control is a critical measure to prevent obesity-related OA. Maintaining a healthy weight and appropriate body fat percentage through a balanced diet and regular physical activity can reduce joint stress.These simple yet effective measures not only mitigate the adverse effects of aging on joint health but also help prevent obesity-induced OA, ultimately improving quality of life and reducing the burden of OA [57].

It is undeniable that this study has some limitations. First, the underperformance of healthcare systems in developing countries leads to an underestimation of OA cases in GBD assessments, along with prevalent misdiagnosis and underdiagnosis. Second, because the original GBD data came from different countries, data quality varied significantly, and the heterogeneity of OA measurements further complicates disease classification and population comparisons. Although GBD employs data cleaning and statistical modeling methods to minimize these limitations, the findings still rely heavily on modeling data, particularly at the national level [58]. Moreover, the lag in GBD data presents another challenge, as current estimates often rely on historical trends, failing to accurately reflect present-day realities. Therefore, strengthening international cooperation, enhancing disease diagnostic standards in underdeveloped countries, and collecting more health data are urgently needed to improve the accuracy of disease burden estimates and research. Despite these limitations, GBD provides critical insights into OA epidemiology and lays a solid foundation for public health policy development and the rational allocation of healthcare resources [59].

To address the limitations outlined and advance osteoarthritis (OA) burden assessment, future studies should prioritize the following directions: 1. Standardization of diagnostic criteria and data collection protocols across low- and middle-income countries to mitigate underdiagnosis and misclassification. 2. Integration of real-time data streams (e.g., electronic health records, wearable sensors) with GBD modeling frameworks to reduce temporal lag and improve responsiveness to emerging risk factors like obesity epidemics. 3. Mechanistic studies on metabolic-inflammatory crosstalk in OA pathogenesis, particularly exploring therapeutic targets. 4. Cost-effectiveness analyses of precision interventions, combining disease-modifying therapies with socioeconomic burden models to guide global resource allocation. These efforts would synergize with the GBD framework to bridge evidence gaps and translate epidemiological insights into actionable clinical strategies.

## 5. Conclusion

In conclusion, as a major public health problem, the global burden of OA is generally on the rise from 1990 to 2021. The highest age-standardised prevalence is located in the high-income Asia-Pacific region, with the lowest in South-East Asia. Countries with higher SDI bear a disproportionate burden of OA, and SDI-related inequalities between countries have increased over time. Overall, women have a higher burden of OA than men. Among the various types of OA, osteoarthritis of the knee had the greatest impact on the overall burden. Together, these findings highlight the enormous pressures on OA prevention and control, such as the increasing number of cases globally, the overall inadequacy of healthcare resources, and inequalities in their distribution. Global health policymakers should consider more targeted interventions and flexible approaches, such as paying more attention to women's health rights and interests, actively assisting less-developed countries, improving the diagnosis and treatment of OA, promoting the hierarchical and rational allocation of healthcare resources, transforming people's health concepts, and meeting the different healthcare needs of each country, so that we can jointly contribute to the prevention, treatment and management of OA.

## Supporting information

**S1 Fig. Trends in the all-age cases and age-standardized incidence and prevalence rates of OA by sex from 1990 to 2021.** Abbreviations: OA = osteoarthritis.
(DOCX)

**S2 Fig. Contribution of different osteoarthritis sites to combined age-standardised prevalence, globally and by GBD region, 2021.** Abbreviations: GBD = Global Burden of Disease.
(DOCX)

**S3 Fig. Frontier analysis based on SDI and OA YLDs in 204 countries and territories.** Abbreviations: SDI = Socio-Demographic Index, OA = osteoarthritis, YLDs = years lived with disability.
(DOCX)

**S4 Fig. The burden of age-standardised YLDs for osteoarthritis attributable to high BMI in 1990 vs 2021.** Abbreviations: YLDs = years lived with disability, BMI = body mass index.
(DOCX)

**S1 Table. The case number of prevalence, incidence and YLDs of OA in 1990 and 2021 for both sexes by GBD regions.** Abbreviations: YLDs = years lived with disability, OA = osteoarthritis, GBD = Global Burden of Disease.
(DOCX)

**S2 Table. Joinpoint regression analysis of the sex-specific age-standardized incidence rate for OA globally from 1990 to 2021.** Abbreviations: OA = osteoarthritis.
(DOCX)

## Acknowledgments

Everyone who contributed significantly to the work has been listed.

## Author contributions

**Data curation:** Yuan Li.

**Formal analysis:** Yulong Li.

**Funding acquisition:** Guihua Tian.

**Investigation:** Yi Lin.

**Software:** Liangqing Huang.

**Writing – original draft:** Xiaoming Xie, Kuayue Zhang.

**Writing – review & editing:** Xinyi Li.

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
