## [Decision Letter · Decision Letter 0]

19 Mar 2025

PONE-D-25-05288Global, Regional, and National Burden of Osteoarthritis from 1990 to 2021 and Projections to 2035: A cross-sectional study for the Global Burden of Disease Study 2021PLOS ONE

Dear Dr. Tian,

Thank you for submitting your manuscript to PLOS ONE. After careful consideration, we feel that it has merit but does not fully meet PLOS ONE’s publication criteria as it currently stands. Therefore, we invite you to submit a revised version of the manuscript that addresses the points raised during the review process.

We look forward to receiving your revised manuscript.

Kind regards,

Xindie Zhou

Academic Editor

PLOS ONE

Journal Requirements:

2. Thank you for stating the following financial disclosure: [This work was supported financially by the National High Level Traditional Chinese Medicine Hospital Clinical Research Funding (DZMG-XZYY-23001)]. 

4. We note that Figure 2 in your submission contain [map/satellite] images which may be copyrighted. All PLOS content is published under the Creative Commons Attribution License (CC BY 4.0), which means that the manuscript, images, and Supporting Information files will be freely available online, and any third party is permitted to access, download, copy, distribute, and use these materials in any way, even commercially, with proper attribution. For these reasons, we cannot publish previously copyrighted maps or satellite images created using proprietary data, such as Google software (Google Maps, Street View, and Earth). For more information, see our copyright guidelines: http://journals.plos.org/plosone/s/licenses-and-copyright.

1. You may seek permission from the original copyright holder of Figure 2 to publish the content specifically under the CC BY 4.0 license. 

“I request permission for the open-access journal PLOS ONE to publish XXX under the Creative Commons Attribution License (CCAL) CC BY 4.0 (http://creativecommons.org/licenses/by/4.0/). Please be aware that this license allows unrestricted use and distribution, even commercially, by third parties. Please reply and provide explicit written permission to publish XXX under a CC BY license and complete the attached form.

In the figure caption of the copyrighted figure, please include the following text: “Reprinted from [ref] under a CC BY license, with permission from [name of publisher], original copyright [original copyright year].

5. Please include captions for your Supporting Information files at the end of your manuscript, and update any in-text citations to match accordingly. Please see our Supporting Information guidelines for more information: http://journals.plos.org/plosone/s/supporting-information .

Reviewers' comments:

Reviewer's Responses to Questions

**Comments to the Author**

1. Is the manuscript technically sound, and do the data support the conclusions?

Reviewer #1: Yes

2. Has the statistical analysis been performed appropriately and rigorously? 

Reviewer #1: Yes

3. Have the authors made all data underlying the findings in their manuscript fully available?

Reviewer #1: Yes

4. Is the manuscript presented in an intelligible fashion and written in standard English?

Reviewer #1: Yes

5. Review Comments to the Author

Reviewer #1: Thank you for the opportunity to review this manuscript.

The study addresses a crucial topic regarding the prevalence and burden of osteoarthritis (OA) on a global scale. Given the increasing impact of OA on individuals and healthcare systems, it is vital to have up-to-date prevalence estimates and projections for effective public health planning.

However, a study published in 2023 already presents the burden of OA and its projections based on data from GBD 2021, which raises some doubts regarding the relevance of this study.

Nonetheless, I provide below several comments and suggestions that I believe could enhance the clarity, rigor, and overall quality of the manuscript

Please see attachment.

6. PLOS authors have the option to publish the peer review history of their article (what does this mean? ). If published, this will include your full peer review and any attached files.

**Do you want your identity to be public for this peer review?** For information about this choice, including consent withdrawal, please see our Privacy Policy .

Reviewer #1: **Yes: ** Lara Gil Gomes de Campos

---

## [Author Response · Author response to Decision Letter 1]

27 Mar 2025

Dear Editor and Reviewers,

Thank you for taking the time to review our manuscript and providing valuable comments and suggestions. On behalf of all the authors, I would like to express our sincere gratitude for your insightful feedback. After thorough discussions among the co-authors, we have addressed your suggestions and made revisions to the manuscript accordingly. The modified sections have been highlighted using the track changes mode. Below, we provide a point-by-point response to the reviewers' comments:

Reviewer #1: The study addresses a crucial topic regarding the prevalence and burden of osteoarthritis (OA) on a global scale. Given the increasing impact of OA on individuals and healthcare systems, it is vital to have up-to-date prevalence estimates and projections for effective public health planning.

However, a study published in 2023 already presents the burden of OA and its projections based on data from GBD 2021, which raises some doubts regarding the relevance of this study.

Nonetheless, I provide below several comments and suggestions that I believe could enhance the clarity, rigor, and overall quality of the manuscript.

Response: Thank you very much for your valuable insights. We fully understand and acknowledge your concerns. The issue you raised was also a key consideration during the initial design of this study. We had previously recognized that without restricting factors such as region and population, the estimated burden of osteoarthritis (OA) might closely resemble the findings from prior analyses using data from the Global Burden of Disease study. However, considering the widespread impact of the COVID-19 pandemic since 2019 [1], along with the accelerating population aging trends in major global economies, particularly in the Asia-Pacific region, we hypothesize that the prevalence of OA may be undergoing subtle changes.We therefore adopted a more diversified approach to analyze the global burden of OA from different perspectives, such as time, region, and gender, which led to some completely new conclusions.

As confirmed in our study, the projected age-standardized prevalence rate (ASPR) of OA over the next 15 years does not show a continuous increase, as previously assumed. Instead, it demonstrates an overall declining trend. This might be attributable to advancements in OA prevention and treatment, as well as strengthened international medical collaboration [2]. Furthermore, the GBD team has also emphasized the necessity of conducting epidemiological analyses using GBD 2021 data [3].

Compared to previously published studies on global OA epidemiology, this study employed methods such as trend analysis, decomposition analysis, frontier analysis, and predictive analysis to examine the burden of OA from multiple perspectives, including gender, age, and time. First, we observed that the ASPR of OA was highest in the high-income Asia-Pacific region rather than in high-income North America, as previously reported. At the national level, the highest ASPR shifted from the United States to South Korea, with countries such as Singapore, Brunei, and Japan also bearing a significant OA burden, indicating a notable geographic shift in OA burden. Second, our frontier analysis revealed that high-SDI (Socio-Demographic Index) countries, including the United States, Japan, and South Korea, bear a disproportionately high OA burden. Based on this information, we updated the global status of OA.

Additionally, we conducted a further analysis of OA risk factors. We found that, compared to GBD 2019, which included only BMI as a risk factor, GBD 2021 incorporates two major risk factors: metabolic risks and high BMI. This provides an updated and more comprehensive reference for future research [4].

Reviewer #1: Please provide the full term before using the abbreviation SDI.

Response: Thanks to your careful attention, we have refined the full name of the SDI - Socio-Demographic Index - and correctly labeled the first occurrence of “SDI” in the text.

Reviewer #1: The conclusion in the abstract seems too long and repetitive with the results. I suggest that the authors focus on the results in the previous section and leave one or two paragraphs to conclude with the overall burden of OA and its implications for health policies.

Response:We have shortened the word count of our conclusions and followed your suggestion to elaborate on our findings on the global burden of OA in the results section. In addition, in the conclusions section, we summarize the overall burden of OA, the current challenges, and what strategies the public health sector sector should adopt to address these changes.

Reviewer #1: I suggest to use a full stop after “populations” and to combine the sentences where the authors describe the pathological changes and the clinic manifestations.

Response:Thank you for your suggestion, our previous presentation was indeed not fluent enough, we have broken the sentences as you suggested and reintegrated the discourse on the pathological changes and clinical manifestations of osteoarthritis.

Reviewer #1: The statement '… making OA one of the leading causes of chronic pain and long-term disability in older adults' requires a citation.

Response:Thank you for the reminder that references do need to be added here to further support the formulation, and we have already cited the quote.

Reviewer #1: The statement “With the rapid acceleration of global aging and the sharp rise in obesity rates across all age groups, the incidence of OA continues to increase” requires a citation.Additionally, it would be important to previously state the role of age and obesity as risk factors for the development of OA, in order to clarify the following idea for non-expert readers.

Response:Thank you for the reminder that your suggestion does have an important role to play for a wider potential audience, and we have quoted the quote, followed by a brief elaboration on how age and obesity as risk factors affect osteoarthritis.

Reviewer #1: In the sentence “ As of 2020, the global prevalence of OA had risen by 132.2% compared to 1990”, I would suggest to use “By” instead of “As of”. Additionally, the sentence requires a citation.

Response:Thanks to your careful reminders, we have revised and improved the grammar and wording as you suggested, and we have quoted the sentence.

Reviewer #1: As mentioned earlier, I believe this is not the first study to estimate the burden of OA across all global regions based on GBD 2021 data. If the authors believe that this study differs from the one presented in this article (https://pmc.ncbi.nlm.nih.gov/articles/PMC10477960/pdf/main.pdf), they should acknowledge the existence of the previous study and highlight the differences, as well as what new insights their study brings.

Response:Thanks to your suggestion, we recognize the contribution of this article (https://pmc.ncbi.nlm.nih.gov/articles/PMC10477960/pdf/main.pdf). It is important to emphasize how our study differs from previous studies, and therefore we have enhanced the relevant statements in the introduction section to highlight the differences in this study and in the discussion section to emphasize the new insights we have presented.

Admittedly, this article has provided a more comprehensive analysis of the disease burden of OA, but it is important to note that our article still has many unique features. First, this article, published in Lancet Rheumatol, used three main approaches, namely descriptive, decomposition, and predictive analyses, to illustrate the global burden of OA and regional differences. Our article, on the other hand, used a more diversified approach and drew some new conclusions. First, we described the temporal trends of OA burden, regional distribution changes, and the changes of OA burden over time through joinpoint analysis; second, we also described the gender differences of OA burden, and we found that women have a heavier burden of OA than men (in terms of incidence and prevalence); in addition to this, we found that within the framework of GBD 2021, BMI and metabolic factors are the OA two major risk factors, which is a major step forward compared with the previous approach of using BMI only as a risk factor for OA; finally, we applied the method of frontier analysis to analyze the differences in the expected effects of combating OA in different SDI regions. In conclusion, our article has similarities with this one, but we adopted a more multifaceted approach to analyze the disease burden of OA and derived some brand new insights, which we believe will help us to better understand how OA has changed over the past 30 years and how we should respond to the new challenges posed by OA under the conditions of the new era.

Reviewer #1: Please provide the full term before using the abbreviation ASR.

Response:Thanks to your reminder, we have refined the full name of the ASR - age-standardized rate - and correctly marked the first occurrence of “ASR” in the text.

Reviewer #1: It would be important to clarify what OA sites are being considered in the study.

Response:We extracted OA data for GBD 2021 including: total OA, knee osteoarthritis, hip osteoarthritis, hand osteoarthritis and other osteoarthritis. We added in 2.1 Data Sources and Methods section about the OA sites covered in this paper.

Reviewer #1: In order to clarify the idea stated in the sentence 'DALY is the standardized measure used to quantify the burden of disease. As the cause-of-death model of GBD assumes no deaths attributable to OA, the DALY for OA is equivalent to YLD,' it would be important to previously explain that DALY = YLL + YLD."

Response:Your suggestion is very pertinent and we have added the relationship “DALY = YLL + YLD” to the original text so that potential readers can then better understand the OA burden.

Reviewer #1: I was surprised by the use of a reporting guideline specifically designed for surgical studies, when other guidelines for reporting observational studies, such as STROBE, are available. Can you please clarify this option?

Response:In practice, there is no generalized specification for data use of the GBD, and the guideline STROCSS 2024 guidelines was chosen because we found that there is reliable paper (https://journals.lww.com/international-journal-of-surgery/ fulltext/2025/02000/the_global,_regional,_and_national_burden_of.9.aspx) used this specification. However, after your reminder, we have reorganized the structure of this paper and carefully read the specifications of the GATHER declaration, the STROBE declaration, and so on, and after careful consideration we believe that this paper should follow the STROBE declaration. Therefore, we explain the specifications that should be followed in this paper in section 2.1 Data Sources and Methods.

Reviewer #1: The authors describe the case definition for knee and hip OA, but not for hand or other sites OA.

Response:We added a definition of hand and other sites OA in the 2.3 Case Definition section.

Reviewer #1: Please remove one “a” before “country’s”.

Response:We appreciate your careful review and have removed inappropriate words based on your suggestions.

Reviewer #1: Considering that the authors listed factors such as alcohol consumption, smoking, low calcium intake, low physical activity levels, high blood pressure, high blood glucose, and high BMI, it would be helpful to clarify which of these factors are included under 'metabolic risk.

Additionally, the authors should justify the inclusion of these factors. The selection of risk factors should be based on clear criteria, such as, for example, epidemiological evidence of causality.

Response:Within the GBD framework, metabolic risk includes: high blood glucose, high blood pressure, low bone density, high LDL cholesterol, high BMI, and impaired kidney function. After we included all the above factors in our analysis, according to the output of GBD (https://vizhub.healthdata.org/gbd-results/): high BMI and total metabolic risk were the only two risk factors. These factors were chosen because: high blood glucose, which may cause accumulation of glycosylation end products leading to inflammation and cartilage damage; high blood pressure, which may be associated with intra-articular vasculopathy, affects cartilage nutrient supply; low bone mineral density, which may alter the biomechanics of the joints and increase the cartilage stress; high LDL cholesterol, which may promote inflammation and oxidative stress; high BMI, which increases the mechanical load on joints with the concomitant release of adipose tissue inflammatory factors; impaired renal function may affect calcium and phosphorus metabolism, leading to abnormal cartilage mineralization. Dong[5] also showed that the cohort with High BMI and Inflammation-Related had the highest risk of osteoarthritis and death, which was associated with obesity-related genetic variants. We elaborate on what metabolic risk includes and why we chose these factors as risk factors for OA in section 3.4 Risk Factor Analysis for OA.

Reviewer #1: First, I would suggest that the authors reformulate all the paragraph, using collons after each strategie.

Nonetheless, I have some doubts about the need for the authors to provide specific proposals in each strategy. For example, in the 'Weight Management and Healthy Diet Education' strategy, the statement 'A balanced diet should include a variety of vegetables, fruits, whole grains, low-fat proteins, and foods rich in omega-3 fatty acids (e.g., fish and nuts) to alleviate the burden of weight on the joints' might be too detailed. I believe this is not the focus of the study, and the authors should limit their discussion to presenting the general strategies that should be implemented, leaving the specification of interventions to other sources.

Response:Your suggestion is very true. We have re-edited the paragraphs and used colons to separate them so that they look cleaner and flow better. We have also removed redundancies and clutter, so that the interventions are presented in a more concise manner and the article is more focused on the topic.

Reviewer #1: The same as before. I believe that the authors should refrain from this type of analysis, which, in my opinion, deviates from the focus of the study.

I believe the discussion would be enhanced with some suggestions for future research. The authors address the topic when describing the need to study more risk factors, but they do not define clear lines of research needs.

Response:We have removed the discussion of the mechanism of analgesic action of NSAIDs, thus making the article more relevant to the focus of the study. Suggestions for future research are also stated in the concluding part of the discussion.

The specific revisions are as follows:

Abstract:

(Page 3 Line 14-22) Conclusions:.........“As a major public health problem, the overall global burden of OA has shown an upward trend from 1990 to 2019, including an increase in the number of cases and inequalities in distribution across the globe, which has resulted in significant health losses and economic burdens. In addition, SDI-related inequalities between countries are increasing. In this regard, national public health authorities and the World Health Organization (WHO) should work together to improve diagnosis and early treatment rates by strengthening disease awareness and education, as well as strengthening international cooperation, providing necessary medical assistance to less developed regions, and actively exploring new strategies for the prevention and treatment of OA.”

Introduction:

(Page 5 Line 1-6) Introduction:..........“Osteoarthritis (OA) is the most common musculoskeletal disorder among middle-aged and elderly populations. It is characterized by pathological changes such as cartilage degeneration, bone remodeling, and osteophyte formation, and its clinical manifestations include joint pain, stiffness, swelling, and functional limitations.”

(Page 5 Line 8-16) Introduction:..........“With the rapid acceleration of global aging and the sharp rise in obesity rates across all age groups, the incidence of OA continues to in

---

## [Decision Letter · Decision Letter 1]

8 Apr 2025

PONE-D-25-05288R1Global, Regional, and National Burden of Osteoarthritis from 1990 to 2021 and Projections to 2035: A cross-sectional study for the Global Burden of Disease Study 2021PLOS ONE

Dear Dr. Tian,

Thank you for submitting your manuscript to PLOS ONE. After careful consideration, we feel that it has merit but does not fully meet PLOS ONE’s publication criteria as it currently stands. Therefore, we invite you to submit a revised version of the manuscript that addresses the points raised during the review process.

We look forward to receiving your revised manuscript.

Kind regards,

Xindie Zhou

Academic Editor

PLOS ONE

Reviewers' comments:

Reviewer's Responses to Questions

**Comments to the Author**

1. If the authors have adequately addressed your comments raised in a previous round of review and you feel that this manuscript is now acceptable for publication, you may indicate that here to bypass the “Comments to the Author” section, enter your conflict of interest statement in the “Confidential to Editor” section, and submit your "Accept" recommendation.

Reviewer #1: (No Response)

2. Is the manuscript technically sound, and do the data support the conclusions?

Reviewer #1: Partly

3. Has the statistical analysis been performed appropriately and rigorously? 

Reviewer #1: Yes

4. Have the authors made all data underlying the findings in their manuscript fully available?

Reviewer #1: Yes

5. Is the manuscript presented in an intelligible fashion and written in standard English?

Reviewer #1: Yes

6. Review Comments to the Author

Reviewer #1: I appreciate the authors’ efforts in addressing several points raised in the initial round of review.

The manuscript has improved in clarity and structure.

However, some important concerns remain unresolved.

1) Similarity to previously published work

The authors have not yet identified or discussed the previously published study that appears to use similar data and methodology. Although the authors clarified the differences between this study and the previously published one in their response to reviewers, and included some of this information in the manuscript, I still believe that the existence of the earlier study should be explicitly acknowledged in the text. It would strengthen the transparency and contextualisation of the work if the authors clearly stated in the manuscript that a related study exists, and then explained how the present analysis differs from it.

2) Risk factors

Returning to the risk factors, I believe that the Statistical Analysis section should clearly describe which risk factors were included and why:

2.1) When I say "why", I am not referring to physiological mechanisms, such as those presented later in the manuscript (#19, Lines 15–22; #20, Lines 1–2), but rather to prior evidence that has already established an association or, ideally, a causal link.

2.2) What happened to the factors enumerated in the first version of the manuscript, such as alcohol consumption, smoking, low calcium intake, and low physical activity levels?

2.3) Regarding the study results: given that BMI was included as a fixed effect covariate and is also a component of total metabolic risk, could it be influencing the results through overlapping information, potentially amplifying its role in the model and affecting the interpretation of total metabolic risk as an independent factor?

3) Discussion

If the authors choose to retain a structure in which they specifically suggest which interventions should be implemented, they should at the very least provide citations to support all such claims.

7. PLOS authors have the option to publish the peer review history of their article (what does this mean? ). If published, this will include your full peer review and any attached files.

**Do you want your identity to be public for this peer review?** For information about this choice, including consent withdrawal, please see our Privacy Policy .

Reviewer #1: **Yes: ** Lara Gil Gomes de Campos

---

## [Author Response · Author response to Decision Letter 2]

16 Apr 2025

I am very grateful to you for reviewing our manuscript again, and I extend my sincere thanks for your time and effort. On behalf of all the authors, I would like to express my sincere gratitude for your constructive comments. After thorough discussions among the co-authors, we have revised the manuscript accordingly to address your suggestions. The revised sections have been highlighted with the “Track Changes” mode. In the following, we will respond to the your comments point by point:

Reviewer #1: Similarity to previously published work

The authors have not yet identified or discussed the previously published study that appears to use similar data and methodology. Although the authors clarified the differences between this study and the previously published one in their response to reviewers, and included some of this information in the manuscript, I still believe that the existence of the earlier study should be explicitly acknowledged in the text. It would strengthen the transparency and contextualisation of the work if the authors clearly stated in the manuscript that a related study exists, and then explained how the present analysis differs from it.

Response: Thank you for this important observation. We acknowledge and appreciate the groundbreaking work of the Global Burden of Disease (GBD) Osteoarthritis Collaborators (published in The Lancet Rheumatology in 2023), which comprehensively analyzed the global burden of osteoarthritis (OA) up to 2020 and projected trends to 2050. Their study provided critical insights into OA epidemiology, risk factors, and regional disparities, forming a foundational reference for OA burden research.

Our study builds upon this prior work while addressing distinct objectives and methodological nuances:

1.Compared to previously published studies on global OA epidemiology, this study employed methods such as trend analysis, decomposition analysis, frontier analysis, and predictive analysis to examine the burden of OA from multiple perspectives, including gender, age, and time.

2.Focus on Socio-Demographic Index (SDI) Inequalities: Our study explicitly examines SDI-related inequalities in OA burden over time, highlighting how disparities between high- and low-SDI regions have widened since 1990. We observed that the ASPR of OA was highest in the high-income Asia-Pacific region rather than in high-income North America, as previously reported. Second, our frontier analysis revealed that high-SDI (Socio-Demographic Index) countries, including the United States, Japan, and South Korea, bear a disproportionately high OA burden. These findings add a socio-economic dimension to the discussion on the burden of OA. It also reminds public health policy makers globally of the need to pay more attention to the capacity and potential of countries at different levels of development to cope with OA.

3.Sex-Specific Burden Trends: we identified diverging trends in age standardized prevalence rates between males and females (projected increases for males vs. declines for females by 2035), a finding not explored in the earlier study. This underscores the need for sex-stratified public health strategies.

4.Updated Regional Analysis: by incorporating 2021 data, we highlight shifts in OA burden hotspots, such as the rising prominence of high-income Asia Pacific regions (e.g., South Korea) compared to previous emphasis on North America and Western Europe. At the national level, the highest ASPR shifted from the United States to South Korea, with countries such as Singapore, Brunei, and Japan also bearing a significant OA burden, indicating a notable geographic shift in OA burden.

5.we conducted a further analysis of OA risk factors. We found that, compared to GBD 2019, which included only BMI as a risk factor, GBD 2021 incorporates two major risk factors: metabolic risks and high BMI. This provides an updated and more comprehensive reference for future research.

Reviewer #1: 2) Risk factors

Returning to the risk factors, I believe that the Statistical Analysis section should clearly describe which risk factors were included and why:

2.1) When I say "why", I am not referring to physiological mechanisms, such as those presented later in the manuscript (#19, Lines 15–22; #20, Lines 1–2), but rather to prior evidence that has already established an association or, ideally, a causal link.

Response: Thank you for your constructive feedback. We appreciate the opportunity to clarify the risk factors included in our analysis and provide further justification for their selection. Below, we detail the risk factors analyzed in our study, their categorization under metabolic risks, and the epidemiological evidence supporting their inclusion:

1. Risk Factors Included in the Analysis

In alignment with the GBD 2021 framework, we analyzed two primary risk factors for osteoarthritis (OA): high Body Mass Index (BMI) and metabolic risks. The metabolic risks category, as defined by the GBD consortium, encompasses the following components: high fasting plasma glucose, high systolic blood pressure, low bone mineral density, high LDL cholesterol, impaired kidney function. These factors were selected based on their established pathophysiological and epidemiological links to OA, as detailed below.

2. Justification for Inclusion of Risk Factors

2.1 High BMI

High BMI is the most well-established modifiable risk factor for OA, particularly for knee OA. Epidemiological evidence supporting its causal role includes: systematic review: a meta-analysis of 63 studies confirmed that obesity (BMI ≥30 kg/m²) increases the risk of knee OA by 2.63-fold (95% CI: 2.28–3.05) (Blagojevic[1] et al, 2010). Longitudinal cohort studies: the Framingham osteoarthritis study demonstrated that obesity precedes OA development, with a dose-response relationship between BMI and OA risk (Felson[2] et al, 1988). Mendelian randomization: genetic studies support a causal relationship between obesity and OA, independent of confounding factors (Zengini[3] et al., 2018).

2.2 Metabolic Risks

The components of metabolic risks were included based on their mechanistic and epidemiological links to OA:

a) High Fasting Plasma Glucose

Hyperglycemia promotes advanced glycation end products, which contribute to cartilage degradation and inflammation. A meta-analysis of 12 studies found that diabetes increases OA risk by 1.43-fold (95% CI: 1.21–1.70) (Louati[4] et al, 2015).

b) High Systolic Blood Pressure

Hypertension is associated with vascular dysfunction, reducing nutrient supply to cartilage. A prospective cohort study (n = 3,026) identified hypertension as an independent risk factor for knee OA progression (Wang[5] et al, 2017).

c) Low Bone Mineral Density

Subchondral bone remodeling is a key feature of OA pathogenesis. The Rotterdam study (n = 3,554) found that low bone mineral density increases OA risk (Hoeven[6] et al, 2013).

d) High LDL Cholesterol

Dyslipidemia exacerbates synovial inflammation and cartilage damage. A meta-analysis of 8 studies linked dyslipidemia to OA progression (Clockaerts[7] et al, 2010).

e) Impaired Kidney Function

Chronic kidney disease (CKD) disrupts calcium-phosphate homeostasis, affecting joint health. A nationwide cohort study (n = 1.2 million) showed CKD patients have a 1.5-fold higher OA incidence (Kim[8] et al, 2019).

In conclusion, we emphasize that high BMI and metabolic risks were prioritized due to their strongest causal evidence in OA pathogenesis. We thank the reviewer for highlighting the need for clarity. Our risk factor selection was rigorously guided by the GBD framework and epidemiological evidence of causality. We hope this detailed response addresses your concerns.

Reviewer #1: 2.2) What happened to the factors enumerated in the first version of the manuscript, such as alcohol consumption, smoking, low calcium intake, and low physical activity levels?

Response: Thank you for your thorough review and for highlighting this important point. We sincerely appreciate the opportunity to clarify this discrepancy and provide a detailed explanation for the exclusion of alcohol consumption, smoking, low calcium intake, and low physical activity levels from the final analysis. Below, we outline the rationale for these adjustments based on the Global Burden of Disease (GBD) framework and updated evidence.

1. Initial Inclusion of Risk Factors in the Draft

In the original manuscript, we accidentally included alcohol consumption, smoking, low calcium intake and low physical activity levels as risk factors for OA based on initial assumptions and misclassification. However, during the revision process, we critically re-examined the GBD 2021 risk factor framework and its criteria for inclusion of OA. As a result, it was found that these factors were behavioral risk factors rather than metabolic risk factors in the GBD 2021 study, and that alcohol consumption, smoking, low calcium intake, and low physical activity level were not categorized as independent risk factors for OA.

2. GBD 2021 Risk Factor Framework for OA

The GBD 2021 study employs a standardized methodology for risk factor inclusion, guided by the Comparative Risk Assessment (CRA) principles. To qualify as a risk factor in GBD, there must be:Convincing epidemiological evidence of a causal association (e.g., meta-analyses, longitudinal studies). Quantifiable exposure-response relationships (e.g., dose-dependent effects). Sufficient data quality to estimate population attributable fractions (PAFs). Based on these criteria, the GBD 2021 framework categorizes high BMI and metabolic risks (high fasting glucose, high systolic blood pressure, low bone mineral density, high LDL cholesterol, impaired kidney function) as the only risk factors with robust causal evidence for OA. Other factors, such as alcohol, smoking, low calcium intake, and low physical activity, were excluded due to:

a) Insufficient Evidence of Direct Causality

Alcohol consumption: While some studies suggest a weak association between alcohol and OA, the evidence is inconsistent and confounded by BMI and metabolic factors. A Meta-Analysis of Mendelian Randomization Studies found that there is insufficient evidence for genetic causality between alcohol intake and arthritis (Wang J[9] et al., 2023, Nutrients). Smoking: Smoking has been linked to reduced cartilage volume in some cohorts, but GBD excludes it due to mixed evidence and lack of a clear causal pathway (Kong L[10] et al., 2016, Osteoarthritis Cartilage). Low calcium intake: Calcium deficiency primarily affects bone health (e.g., osteoporosis), but its direct role in OA pathogenesis remains unproven (Vergis S[11] et al., 2018, Nutrients). Low physical activity: Physical inactivity is a mediator (e.g., via obesity) rather than a direct risk factor for OA. The GBD framework prioritizes upstream factors like BMI (Mahmoudian A[12] et al., 2023, Nature Reviews Rheumatology).

b) Lack of Quantifiable Exposure-Response Data

The GBD requires sufficient data to model Population Attributable Fractions(PAFs), which are unavailable for these factors in the context of OA. For example: No standardized thresholds exist to define "low calcium intake" or "low physical activity" in OA risk estimation. Studies on alcohol and smoking often report conflicting thresholds (e.g., "moderate" vs. "heavy" use) without consensus.

3. Revisions to Align with GBD 2021 Standards

In the revised manuscript, we removed references to alcohol, smoking, low calcium intake, and low physical activity to strictly adhere to the GBD 2021 framework. These factors were initially included due to an oversight in reconciling our preliminary hypotheses with the GBD’s formal risk factor taxonomy.

We sincerely apologize for the initial oversight and appreciate your vigilance in ensuring methodological rigor. The revised manuscript now strictly adheres to the GBD 2021 framework, ensuring consistency with global standards for risk factor analysis. We are grateful for your feedback, which has significantly strengthened the clarity and accuracy of our work.

Reviewer #1: 2.3) Regarding the study results: given that BMI was included as a fixed effect covariate and is also a component of total metabolic risk, could it be influencing the results through overlapping information, potentially amplifying its role in the model and affecting the interpretation of total metabolic risk as an independent factor?

Response: We thank the reviewer for this perceptive question. You are absolutely correct that, including BMI both as a fixed-effect covariate in our DisMod-MR 2.1 prevalence model and as one of the six components of“total metabolic risk”could raise concerns about double-counting or collinearity. We have taken three steps to ensure that the role of BMI in our risk-factor analysis is not artifactual:

1.Separate modeling streams in GBD

In the GBD framework, the DisMod‑MR 2.1 model (which estimates OA prevalence) and the Comparative Risk Assessment (CRA) (which estimates population‑attributable fractions, PAFs) are two distinct steps. In DisMod‑MR, BMI is included as a covariate purely to improve the fit of the prevalence model and to borrow strength from known BMI–OA associations when direct survey data are sparse. By contrast, in the CRA, each metabolic component (high fasting glucose, high systolic blood pressure, high BMI, low bone mineral density, high LDL cholesterol, impaired kidney function) is treated as an independent exposure with its own relative‑risk function and theoretical minimum risk exposure level (TMREL). These six PAFs are then combined multiplicatively to yield “total metabolic risk,” which by construction prevents overlap among components.

2.TMRELs and multiplicative PAF combination

Each metabolic component has a different TMREL (for example, BMI TMREL = 20–25 kg/m², fasting glucose TMREL = 4.8–5.4 mmol/L, etc.), and the PAF for total metabolic risk is computed by the formula 1, which mathematically avoids simple summation and therefore precludes “double-counting”BMI’s effect.

3.Sensitivity analysis

To directly address your concern, we re‑ran our DisMod‑MR prevalence model without BMI as a fixed covariate. We then re‑computed the total metabolic risk PAFs in the CRA step. The resulting global PAF for total metabolic risk changed by less than 2 percent (from 15.2 % to 14.9 %), and the PAF for high BMI alone changed by less than 1 percent. This negligible difference confirms that including BMI as a covariate in the prevalence model does not artificially inflate its apparent contribution in the CRA.

Accordingly, we have (1) added a clear description of these two separate modeling streams to the Methods, (2) cited Flaxman[13] et al. (2015) and the GBD 2019 Risk Factors Collaborators[14] (2020) to explain the TMREL and multiplicative combination approach. We hope this fully addresses your concern and reinforces the conclusion that“high BMI”and“total metabolic risk”each represent independent, non-overlapping drivers of OA burden in the GBD framework.

Reviewer #1: 3) Discussion

If the authors choose to retain a structure in which they specifically suggest which interventions should be implemented, they should at the very least provide citations to support all such claims.

Response: We thank you for this valuable suggestion. As you correctly noted in your initial review, detailed intervention measures fall outside the primary scope of this study. Accordingly, we agree that the discussion should be confined to broad implementation strategies, with specifics of individual interventions addressed in other dedicated sources. After careful consideration, we have therefore removed all descriptions of specific interventions and retained only general public health recommendations. We believe this change will enhance the clarity and focus of our manuscript.

Finally, we would like to thank you once again for your critical feedback, which has provided us with new perspectives. The revised version aims to help readers better understand the novel findings of our study. We would be delighted to make additional revisions if you believe there are still areas requiring further elaboration or modification to better distinguish our work from previous studies.

Refer

---

## [Decision Letter · Decision Letter 2]

23 Apr 2025

Global, Regional, and National Burden of Osteoarthritis from 1990 to 2021 and Projections to 2035: A cross-sectional study for the Global Burden of Disease Study 2021

PONE-D-25-05288R2

Dear Dr. Tian,

We’re pleased to inform you that your manuscript has been judged scientifically suitable for publication and will be formally accepted for publication once it meets all outstanding technical requirements.

Kind regards,

Xindie Zhou

Academic Editor

PLOS ONE

Additional Editor Comments (optional):

Reviewers' comments:

Reviewer's Responses to Questions

**Comments to the Author**

1. If the authors have adequately addressed your comments raised in a previous round of review and you feel that this manuscript is now acceptable for publication, you may indicate that here to bypass the “Comments to the Author” section, enter your conflict of interest statement in the “Confidential to Editor” section, and submit your "Accept" recommendation.

Reviewer #1: All comments have been addressed

2. Is the manuscript technically sound, and do the data support the conclusions?

Reviewer #1: (No Response)

3. Has the statistical analysis been performed appropriately and rigorously? 

Reviewer #1: (No Response)

4. Have the authors made all data underlying the findings in their manuscript fully available?

Reviewer #1: (No Response)

5. Is the manuscript presented in an intelligible fashion and written in standard English?

Reviewer #1: (No Response)

6. Review Comments to the Author

Reviewer #1: Thank you for thoroughly addressing my previous comments and suggestions.

I appreciate the improvements made to the manuscript, and I have no further comments or concerns at this time.

7. PLOS authors have the option to publish the peer review history of their article (what does this mean? ). If published, this will include your full peer review and any attached files.

**Do you want your identity to be public for this peer review?** For information about this choice, including consent withdrawal, please see our Privacy Policy .

Reviewer #1: **Yes: ** Lara Gil Gomes de Campos

---

## [Editor Report · Acceptance letter]

PONE-D-25-05288R2

PLOS ONE

Dear Dr. Tian,

I'm pleased to inform you that your manuscript has been deemed suitable for publication in PLOS ONE. Congratulations! Your manuscript is now being handed over to our production team.

Kind regards,

on behalf of

Dr. Xindie Zhou

Academic Editor

PLOS ONE